# Thinking Outside the Bug: Molecular Targets and Strategies to Overcome Antibiotic Resistance

**DOI:** 10.3390/ijms20061255

**Published:** 2019-03-13

**Authors:** Ana Monserrat-Martinez, Yann Gambin, Emma Sierecki

**Affiliations:** 1European Molecular Biology Laboratory Australia (EMBL Australia) Node in Single Molecule Science, Sydney, NSW 2031, Australia; a.monserrat-martinez@unsw.edu.au; 2School of Medical Sciences, The University of New South Wales, Sydney, NSW 2031, Australia

**Keywords:** antibiotic resistance, host–pathogen interactions, bacterial effectors, protein–protein interactions, antimicrobial peptides and proteins

## Abstract

Since their discovery in the early 20th century, antibiotics have been used as the primary weapon against bacterial infections. Due to their prophylactic effect, they are also used as part of the cocktail of drugs given to treat complex diseases such as cancer or during surgery, in order to prevent infection. This has resulted in a decrease of mortality from infectious diseases and an increase in life expectancy in the last 100 years. However, as a consequence of administering antibiotics broadly to the population and sometimes misusing them, antibiotic-resistant bacteria have appeared. The emergence of resistant strains is a global health threat to humanity. Highly-resistant bacteria like *Staphylococcus aureus* (methicillin-resistant) or *Enterococcus faecium* (vancomycin-resistant) have led to complications in intensive care units, increasing medical costs and putting patient lives at risk. The appearance of these resistant strains together with the difficulty in finding new antimicrobials has alarmed the scientific community. Most of the strategies currently employed to develop new antibiotics point towards novel approaches for drug design based on prodrugs or rational design of new molecules. However, targeting crucial bacterial processes by these means will keep creating evolutionary pressure towards drug resistance. In this review, we discuss antibiotic resistance and new options for antibiotic discovery, focusing in particular on new alternatives aiming to disarm the bacteria or empower the host to avoid disease onset.

## 1. Introduction

During the golden age of antibiotic discovery (1940–1970) [1], multiple natural products with antimicrobial properties were discovered. These molecules target essential bacterial functions or growth processes, mainly aiming to interfere with or destroy the bacterial cell wall, the cell membrane or crucial bacterial enzymes involved in nucleic acid and protein synthesis. These “first generation antibiotics” all function by targeting structures unique to bacteria and not present in mammals to minimize side effects. Although they were extremely useful at first, microorganisms quickly evolved and became resistant, making some of these antibiotics obsolete. As such, a need for new antibiotics appeared, rekindling the field of antibiotic and microbial research.

The first time that a bacterial strain resistant to penicillin appeared was in 1947. Four penicillin-resistant staphylococci strains were isolated from patients within a year of penicillin introduction in the clinic, two years after becoming widely available, and not even 20 years from its discovery in 1928 [2,3]. In 1959, the pharmaceutical industry developed methicillin, capable of avoiding penicillinase, the enzyme which breaks penicillin rings. Methicillin reached the market in 1960; however, *Staphylococcus aureus* strains resistant to this antibiotic appeared only one year later, proving how fast bacteria can evolve and become resistant [4,5]. 

Over the next 20 years, pharmaceutical companies actively worked towards the discovery and the development of new antibiotics: ampicillin, several cephalosporins, vancomycin, and levofloxacin were all discovered before year 2000 [6,7,8,9,10]. Nowadays, the number of active programs looking for new antibiotics is scarce. In 2014, it was estimated that among the large pharmaceutical companies, only four had active antibiotic discovery programs, down from 18 in 1990 [11]. This is due to the high difficulty and cost of finding new antibiotics. According to a study by GlaxoSmithKline that summarized target-based drug research against *Haemophilus influenzae* [12], from 70 screenings of libraries containing between 260,000 and 530,000 molecules, only five candidates showed good results as a potential antibiotic, none of them passing further clinical trials to become licensed [1]. For the programs still active, the discovery of compounds from natural sources focuses mainly on fungi and soil bacteria screening [13]. Alternatives like the modification of the classic antibiotics using molecular biology and chemistry tools to find new ways to overcome resistance [14] are under intense study, as they are often more successful than screening for new compounds.

In this review, we explain some of the most common chemistry tools exploited since bacterial resistance appeared and discuss new alternatives based on a change of target to fight pathogen infections.

## 2. Bacterial Resistance and Evolution of Antibiotics

The word “resistance” relates to the ability of bacteria to survive a specific antibiotic treatment. Some bacterial species are naturally resistant to a given group of antibiotics. Acquired resistance means that only some strains from a specific species are resistant to an antibiotic, but not the whole species. This resistance can appear due to a spontaneous mutation in the chromosomal DNA or can happen extra-chromosomally, such as when bacteria exchange plasmids or transposons. Some of the most common resistance mechanisms include modification/inactivation of the antibiotic itself, changes to the external membrane permeability, the appearance of efflux pumps and changes to the bacterial target site [15,16,17].

Several approaches have been developed to fight the problems with current and emerging bacterial resistance. Some of these approaches focus on targeting the same sites as 1st generation antibiotics (bacterial cell wall, the cell membrane or essential bacterial enzymes) with chemically modified antibiotics or with a combination of several antibiotics. Some 2nd, 3rd or 4th generation antibiotics are such modified compounds with improved pharmacological properties but with the same mechanism of action. The main drawback to this approach is that old target sites are usually directly related to essential bacterial processes. This creates a strong adaptation pressure: bacteria will try to readjust to the new environment for its survival. Those individuals with the greater capacity to produce genetic variability will have the greatest potential of finding a way to overcome the effect of the antibiotic, leading to the appearance of resistance. In cases where the resistance appears due to mutations in the target site or through the development of efflux pumps which remove the antibiotic out of the bacteria, the problem becomes extremely challenging, as the new analogs are also likely to be affected. 

The use of combinations of single-target antimicrobials is also a standard therapy against some infections like *Helicobacter pylori* or *Mycobacterium tuberculosis* [18]. This is a recommended strategy when the compounds show synergy or when the aim is to target different pathogen subpopulations. Two or more drugs administered simultaneously are also used in empiric treatments (treatment before the pathogen is identified) in intensive care units to cover a wide range of the bacterial spectrum [18], both to prevent and overcome the emergence of resistant pathogens.

Another approach is to find novel mechanisms or novel target sites in bacteria. The search for novel unexploited targets has been the main strategy of the scientific community for years. When selecting a new target, different criteria have to be met: it has to be present in a specific spectrum of bacteria, it has to be druggable and it must not be present in humans in any homolog form. Once the therapeutic target is identified and validated by demonstrating that affecting the target will have a direct bactericidal or bacteriostatic effect, the next step is to find a molecule that is effective against that target and safe enough for use. For example, an enzyme inhibitor or a molecule that interferes with binding to the target.

### 2.1. First Generation Antibiotics and Their Analogs

The vast majority of antibiotics present in the market today are improved versions of classic antibiotics derived from natural products [1]. Modifications to add new intramolecular contacts and/or to increase the number of drug–enzyme interactions can be used to lower the potential appearance of resistance [18]. Chemical modification of existing drugs has proven to be the most efficient way to develop novel drugs against resistant strains [1]. This is the case for the newest vancomycin analogs. Vancomycin is a glycopeptide antibiotic produced by *Nocardia orientalis*. It interrupts bacterial cell wall synthesis by binding with high affinity to peptidoglycan precursors ending with two copies of d-Alanine (d-Ala) [19]. Vancomycin prevents cell wall assembly and makes the total cell wall content decrease during the time of antibiotic treatment [20,21]. After approximately 30 years of clinical use, the first vancomycin-resistant bacteria (VRE, vancomycin resistant enterococcus) were detected [22,23,24]. A cassette of genes carried chromosomally or extra-chromosomally on a plasmid, called the *Van* operon [25], gives the bacteria the ability to cleave the d-Ala-d-Ala repeats necessary for antibiotic interaction. New vancomycin analogs were produced to overcome this problem [26,27], but bacteria kept evolving and soon became resistant again (Figure 1). Other treatment options, such as linezolid and daptomycin, have been shown to rapidly provoke resistance as well [28,29]. In 2017, a new vancomycin analog was produced [30]. This molecule is supposed to be a durable antibiotic with special features (discussed in Section 2.3) to avoid the appearance of resistance. However, to be able to produce it, remodeling of vancomycin was not sufficient and dual ligand binding techniques had to be used. 

Another good example is rifamycin SV, an antibiotic belonging to the family of ansamycin antibiotics [31,32]. The precursor, rifamycin B, is found in *Amycolatopsis mediterranei* ferment when it grows in the presence of diethylbarbituric acid, and it is poorly active [33]. Rifamycin B can be chemically modified (by oxidation, hydrolization, and reduction) to produce rifamycin SV [34,35]. Rifamycin SV binds to the DNA-dependent RNA polymerase of prokaryotes [36,37]. The main way pathogens develop resistance to rifamycin is by modifications to the drug target, rpoB [38,39]. For example, *Mycobacterium tuberculosis* mutates at a rate of 10^−8^ to 10^−9^ mutations per bacterium per cell division when exposed to the related antibiotic rifampin [40] making Rifamycin SV of use only in drug combinations or in case of clinical emergencies [41,42]. 

### 2.2. Prodrugs and Antibiotic Derivatives

Prodrugs are inactive derivatives of a known active molecule that have been chemically modified to temporarily alter their pharmacological activity. Properties like bioavailability, absorption, half-life or permeability can be modified by the addition or removal of parts of the molecule without compromising its antimicrobial activity [43]. The prodrug then requires transformation within the body in order to produce the active drug.

One of the first successful prodrugs developed was pivampicillin [44], a derivative of ampicillin which has been modified to increase its absorption rate (Figure 2A). Once the molecule is absorbed in the gastrointestinal tract, it is enzymatically hydrolyzed and ampicillin (together with pivalic acid) is released. The use of pivampicillin produces serum concentrations of 2–3 times the expected value in humans when compared to equivalent doses of ampicillin [45,46,47].

The fusion of active components by a linker is another alternative. Two different molecules (i.e., two antibiotics or one antibiotic together with another drug) are bound together and are absorbed as one. Hydrolysis of the molecule releases the two active compounds that are then able to perform their role. Ampicillin, for example, or β-lactam antibiotics in general, have been combined with β-lactamase inhibitors, small molecules that inhibit β-lactamases, the enzyme that breaks β-lactam antibiotics [48]. The resulting compounds have several benefits. Improved, simultaneous, and constant absorption ratios [49] makes these molecules enticing starting points for further modifications. Such is the case of sultamicillin, a broad-spectrum antibiotic used as empiric therapy in intra-abdominal, gynecologic or skin infections [50,51,52,53,54]. Sultamicillin links ampicillin and sulbactam by a methylene group, enhancing ampicillin’s microbial activity by 4- to 32-fold [55] (Figure 2B). This option is sometimes preferred compared to “simple” prodrugs, as it avoids therapeutically inactive compound release (i.e., pivalic acid, in the abovementioned example).

### 2.3. Discovering New Molecules Using Bioinformatics and Rational Design

Rational design is a tool used to create new molecules with specific characteristics based on how the molecule’s structure will affect its behavior. Thanks to bioinformatics, we can now study large compound libraries and extract relevant information about how minor changes in a molecule can affect its performance and understand the structure–activity relationship. The data can then be filtered to create virtual models to satisfy a given functionality (molecule size, overall charge and hydrophobicity, binding to a specific target, etc.) and combine them to design a molecule.

Rational design became popular in drug development since its first instances in the 1950s [56]. It can be applied broadly: from designing molecules that can target a specific pocket in an enzyme to creating drugs that hijack a protein from a metabolic pathway. This approach requires a deep understanding of the biology of the bacteria and a vast knowledge of the structure of the target. 

A perfect example for rational design is linezolid, an alternative treatment for vancomycin-resistant bacteria. Linezolid is a completely de novo synthetic drug: it does not occur in nature and it was not developed by building upon a naturally occurring skeleton [57,58]. While most protein synthesis inhibitors target the elongation phase [59], linezolid is a protein synthesis inhibitor that inhibits the initiation phase of the bacterial protein synthesis process [60] (Figure 3), providing a new way to overcome the appearance of resistance. 

Another good example of rational design is the new vancomycin analog designed in 2017. As mentioned in Section 2.1, the naturally produced vancomycin molecule was modified initially to fight the resistant strains that appeared over time. Binding pocket modifications were designed to provide both d-Ala-d-Ala and d-Ala-d-Lactate binding to overcome the molecular basis of vancomycin resistance (vancomycin only binds to d-Ala-d-Ala). However, Okano et al. [30] discovered that the addition of a quaternary ammonium salt and (4-chlorobiphenyl)methyl (CBP), both in C-terminal position, provide a 200-fold increase in antimicrobial potency of the vancomycin analog. The addition of the ammonium salt induces cell wall permeability, a mechanism of action completely independent of the d-Ala-d-Ala/d-Ala-d-Lactate binding. Adding CBP inhibits cell wall biosynthesis by directly inhibiting transglycosylase, another mechanism independent of d-Ala-d-Ala/d-Ala-d-Lactate binding. With these peripheral changes, vancomycin acts now through three antimicrobial mechanisms, making resistance less likely to appear [30].

The rational design of antibiotics (both de novo and hybrids of several antimicrobial pharmacophores) display new or additional drug target binding sites that increase antimicrobial effect in a synergistic manner [30] and decrease resistance susceptibility [59].

### 2.4. Combinatorial Chemistry and High Throughput Screening

The synthesis of molecules in a combinatorial way can quickly produce a vast number of molecules [61]. Combinatorial chemistry involves the rapid synthesis of molecules or materials that are structurally related. Hundreds of compounds sharing a common skeleton are produced based on systematically substituting the different moieties present in a molecule and observing how each change modifies its activity [62,63,64].

Despite the huge potential of the technique, the pharmaceutical industry only started to implement this tool in the 1990s [65,66,67]. Nowadays, virtual libraries of compounds are created with millions of molecules containing all the possible structures and combinations available for a given pharmacophore [68]. Computer programs then analyze the library and choose the best hits based on specific requirements set by the researcher. Then, the selected molecules are then synthetically produced to determine their physico-chemical properties (pharmacokinetics, pharmacodynamics, cell entry, toxicity, etc.). 

In the best-case scenario, a lead compound is found. This molecule is then studied in depth to obtain a candidate that will undergo the final validation studies (more in vitro and in vivo studies) prior to the preclinical phase (Figure 4). 

The majority of the compounds that originally started in the screening have to be dropped, as it is extremely difficult to fulfil the pharmacological characteristics (solubility, toxicity, sensitivity, etc.) demanded by world health organizations without losing target specificity or affinity. It is estimated that only 250 molecules out of 10,000 reach preclinical trials [69].

Screenings are particularly powerful when combined with rational design, where molecules targeting a specific active site or pathway can be produced and then virtually screened. The pharmaceutical industry produces more than 100,000 new compounds per year that are screened [70], seeking for new lead compounds useful to treat any diseases. 

### 2.5. Towards New Approaches for Antibacterial Drug Design

All the different approaches described above are based on the discovery or modification of molecules targeting structures and processes essential for bacterial survival, such as cell wall biosynthesis, cell membranes, type II topoisomerases, ribosomes, transcription, and folate biosynthesis [71,72]. In a situation of stress or a threat, like the presence of an antibiotic, bacteria with greater capacity to produce genetic variability will stand out in terms of survival [73]. The ability of bacteria to evolve and adapt, by mutation or by horizontal gene transfer is greater than in the majority of organisms [74,75], and evolutionary pressure will therefore select for drug-resistant individuals.

Antibiotics, together with natural selection, favor the survival of the mutant bacterial cells that can adapt to these conditions. Antibacterial agents that target single enzymes/molecules essential for the bacteria are subject to the greatest selection pressure and result in the development of high-level resistance resulting from single-step mutations in the target molecule, like the case of rifamycin or vancomycin from Section 2.1.

However, the advantage of antibiotic resistance can bring with it some downsides like a reduction in fitness under non-stressed conditions [76], (i.e., slower reproduction [77]). Such is the case of ciprofloxacin, an antibiotic from the quinolone family, and *Salmonella* species. Ciprofloxacin binds to the DNA gyrase of the bacteria inhibiting bacterial replication [78]. In the last 10 years, the frequency of strains resistant to ciprofloxacin has increased [79]. This resistance has appeared due to the accumulation of mutations in the DNA gyrase [80]. However, these mutations make the gyrase less efficient in its role of reducing the molecular tension of the DNA strands during DNA replication, making the resistant bacteria slower growing compared to wildtype.

Targeting processes that are not crucial for bacterial survival or taking advantage of the weak points of resistant bacteria opens up a whole new umbrella of possibilities for the development of antibacterial drugs. When non-essential bacterial processes, not compromising bacterial growth or survival, are targeted, the decreased evolutionary pressure for adaptation should reduce the development of drug resistance [81]. 

## 3. Targeting Non-Essential Processes: The Key to Overcome Resistance

Interactions between microbes and the hosts they colonize are central to both health and disease. Some of these interactions have developed over time as a result of an evolutionary co-existence and adaptation [82]. Bacterial proteins, from symbiotic or pathogenic bacteria, interact with cell receptors of host cells, create pores in the membranes, modify the host cell environment (i.e., toxins), affect cell longevity, and stimulate different cell pathways during infection, among other functions [83,84,85,86,87,88]. They can cause an extensive rewiring of the host cell signaling network, creating a high impact on its health. 

Understanding the protein interactions happening during an infection provides a whole new world of options to target non-essential processes like bacterial adhesion, communication or host hijacking. In this section, we will describe in detail some non-essential bacterial processes that can be targeted and explore the different ways by which protein–protein interactions can be used as a new weapon against bacterial infections. 

### 3.1. Bacterial Communication: Quorum Sensing

In bacteria, chemical communication involves producing, releasing, detecting, and responding to small hormone-like molecules called autoinducers [89]. The information supplied by these molecules is critical for synchronizing the activities of large groups of cells. The communication can be intra-species, inter-species or inter-kingdom. Most of these interactions probably evolved to allow co-existence of a bacterium with its host [81].

One of the main mechanisms of bacterial communication is Quorum sensing (QS). With QS, bacteria can sense the environment and other microbial communities (and their density) and use that information to their interest: when enough bacteria are present, the concentration of autoinducers reaches a threshold that allows the bacteria to sense a critical cell mass, and in response, to activate or repress gene expression [90] (Figure 5A). Bacteria can minimize host immune responses by delaying the production of virulence factors until sufficient bacteria have amassed and are prepared to overcome host defense mechanisms and establish infection [91]. Quorum sensing also gives bacteria the ability to migrate to a more suitable environment or to form biofilms if required. In general, QS gives the bacteria the ability to act as a multicellular organism. 

Biofilm formation is a process whereby microorganisms attach and grow on a surface and produce extracellular polymers that facilitate attachment and matrix formation. Biofilms can be formed at the sites of implanted medical devices as implants or catheters [91]. Formation of biofilms around the devices and encrustation is a clinical complication that can threaten the patient’s life. To reduce these infections, systemic antibiotic prophylaxis as well as local administration of antimicrobial agents are administered [1]. Some alternatives being studied are also coating the device with antibacterial compounds [81] or making the implant material resistant to colonization by physiochemical modification of the biomaterial surface to create anti-adhesive surfaces. This last approach, together with the inhibition of cell-to-cell signaling prior to biofilm formation, is especially interesting as it does not rely on the use of traditional antibiotics.

Increasing our understanding of the chemical signaling among bacterial cells can lead to new drugs preventing the recognition of autoinducers. Drugs targeting peptides involved in QS are specific for each species, as QS systems vary among them. Gram-negative bacteria like *Pseudomonas aeruginosa* depend mainly on derivatives of acyl-homoserine lactones (AHLs) [81,92,93] or other molecules that are synthesized from S-adenosylmethionine (SAM) [94]. These molecules are able to diffuse freely through the bacterial membrane. AHLs are produced by LuxI-type synthases, present in hundreds of bacterial species [95,96]. Acyl-homoserine lactones bind to LuxR-type receptors, transcription factors located in the cytoplasm which contain a N-terminal ligand-binding domain and a C-terminal DNA-binding domain [97]. Upon binding the AHL ligand, the ligand–receptor complex binds to DNA and elicits a change in gene transcription [98,99] (Figure 5B). Targeting the synthesis or binding of AHL could interfere with *P. aeruginosa* signaling and QS establishment.

Gram-positive bacteria typically use oligopeptides as signaling molecules for QS. Oligopeptides are secreted by transporters, as they are impermeable to biological membranes. As receptors, oligopeptides use a two-component system consisting of a membrane-bound histidine kinase receptor and a cytoplasmic response regulator, which acts as a transcription factor, regulating gene expression [100,101] (Figure 5C). Gram-positive oligopeptide autoinducers are not variations on a single core molecule, like in the case of Gram-negative bacteria; the oligopeptides are genetically encoded, and thus each species of bacteria is capable of producing a peptide signal with a unique sequence [102]. In the case of *S. aureus*, four different propeptides, that become active autoinducer peptides (AIPs) after processing by truncation and cyclization, are produced [102,103]. Once active, they are able to bind to the histidine kinase receptor. The produced AIPs are seven to nine residues in length and possess a five-membered ring: this structure is critical for activity [104] (Figure 6). The sequence of AIPs is highly variable, resulting in four specificity groups in *S. aureus* and more than 25 in other staphylococci strains [104,105,106,107]. The kinase receptor, the response receptor and the rest of molecules interacting with the peptides, vary in order to maintain the specificity of the interaction [105,108,109]. Additionally, there is ligand-mediated signaling cross-talk: most AIPs activate their native receptor while competitively inhibiting the activation of the other receptors [105,107,110]. The *S. aureus* two-component quorum sensing system highlights the importance of proteins and peptides in bacterial communication and the high versatility of these molecules to control precisely complex signaling processes within the cell.

As QS and biofilm production do not control processes essential for cellular survival [81], the creation of peptides able to interact with and hijack the bacterial autoinducers is a feasible approach that would provide drugs with narrow spectra and mild selective pressure towards the development of resistance. By hijacking the autoinducers, cell communication is avoided, and disease on-set could be averted or at least slowed down [101]. 

Enzymatic and non-enzymatic inactivation of QS signals is being studied and exploited for QS inhibition. Degradation of AHL or its precursor SAM, can be achieved by lactonases, acylases or oxidoreductases [101,111,112,113,114]. Acyl-homoserine lactones signaling inactivation was first observed in 2000 by Dong et al. [102]. They found a lactonase that inactivates the AHL quorum sensing signal and attenuates the virulence of *Erwinia carotovora*. This lactonase had no significant homology with any known sequence with the exception of a HXHXD sequence motif shared with metallo-β-lactamase [112]. A second AHL-inactivating lactonase was discovered in 2002 and shares less than 25% identity with the first one described [115]. Importantly, the conserved HXHD motif was present, proving once again the importance of molecular interactions and small motifs in controlling and hijacking bacterial processes.

The development of monoclonal antibodies and the creation of peptides in which the length or sequence of AIPs are varied by rational design [100,116] is also being studied with interesting outputs. In 2000, Lyon et al. [117] discovered that modifications in the AIP tail might lead to the generation of global inhibitors of the *agr* response in *S. aureus*, and in 2007 the discovery of an anti-autoinducer monoclonal antibody against AIP-IV of *S. aureus* that efficiently inhibits QS was reported [118].

### 3.2. Host–Pathogen Interactions

Host–pathogen interactions are defined by how microbes or viruses sustain themselves within host organisms on a molecular, cellular, organismal or population level [119]. Pathogenicity is the ability of an organism to cause disease and harm the host. It is a qualitative term and depends on the microbial ecosystem: some organisms will only be pathogenic under certain circumstances, such as in the case of opportunistic pathogens or infections in immunocompromised patients.

The pathogenicity of an organism ultimately depends on its DNA, both chromosomal and acquired in the form of plasmids. Some bacteria have genes encoding for toxins, proteins involved in cell adhesion or colonization. In this section, we will discuss how cell adhesion, virulence factors, and signaling can be targeted to avoid disease onset.

#### 3.2.1. Cell Adhesion and Colonization

The establishment of infection by bacterial pathogens requires adhesion to host cells, colonization of tissues, and in some cases, cellular invasion. Cell adhesion requires an intimate relationship between bacterial and host proteins, where the interaction with the cell’s receptors is key. 

Anchoring to the host cell membrane by cell receptors is one of the first steps towards disease onset, especially for Gram-negative bacteria which rely on secretion systems for infection. Secretion systems are protein complexes present on the cell membranes of bacteria used to inject proteins to assist in the infection of host cells. They are essential for some pathogens to manipulate the host and establish a replicative niche [120]. In general, they are conserved and have many similarities among them. Several different classes of bacterial secretion systems have been identified, based on activity, structure, and composition. So far, eight different types have been described for Gram-negative bacteria, four for Gram-positive, and two common to both types [120].

Some pathogenic *Yersinia* spp. depend on the type three secretion system (T3SS) to avoid phagocytosis and resist the host primary immune defense after infection onset [121]. Type three secretion systems are structurally similar to the basal body of bacterial flagella. They span both the inner membrane and the outer membrane of the bacteria. A needle-like structure is projected from the bacterial cytoplasm into the host plasma membrane, physically connecting them and creating a pore to inject bacterial effector proteins into the host cell. Once *Yersinia* is in the host organism, it uses a T3SS to anchor itself to macrophages and injects YopH into the cytosol, a protein with tyrosine phosphatase activity [122,123]. YopH dephosphorylates several macrophage proteins, reducing their phagocytic ability [124].

Other bacteria like *Shigella*, depend on the T3SS for their uptake into the host cell. *S. flexneri* uses the T3SS to secrete IpaA, IpaB, IpaC, and IpaD, four proteins directly involved in bacterial invasiveness [125,126,127]. These proteins stimulate the clustering of host cell integrins and the phosphorylation of tyrosine kinases of host cytoskeletal proteins, both signals leading ultimately to bacterial internalization [128,129].

In some Gram-negative bacteria, pathogenicity is directly linked to the capacity of the bacteria to deliver effector proteins into host epithelia to alter cellular physiology. This is prevented if the T3SS (or any other secretion system) does not interact with the plasma membrane of the host cell. Targeting the interaction between the proteins of the secretions systems and the host cells could be a valid strategy to prevent different bacterial infections, as many species of *Salmonella, Shigella*, *Yersinia* or *Pseudomonas* [130] infect cells through secretion systems. In this case, the likelihood of resistance developing is minimal, as although secretion systems are essential for bacterial invasion, they are not crucial for bacterial survival. Moreover, targeting processes where host proteins are involved makes this therapeutic option enticing as they are relatively stable targets, since they are under less evolutionary pressure due to the lower reproduction capacity of the host cell. 

Active and passive immunization against secretion system components has been studied before. Experiments using antisera against *P. aeruginosa*’s secreted protein PcrV, one of the three secreted proteins forming the T3SS, showed inhibition of the hemolytic activity and of the translocation of exotoxins into mammalian cells [131,132]. In 2009, a monoclonal antibody against PcrV was engineered [133]. The antibody acts at the level of the formation of the translocation pore in membranes of infected host cells by blocking the function of PcrV. When this essential component of the apparatus is blocked, the T3SS cannot inject toxins into the host cell.

Another common structure crucial for adhesion both in Gram-negative and Gram-positive bacteria are pili. The recent discovery of the mechanisms by which pili assemble in Gram-positive bacteria, showed that pili proteins are sorted by a conserved cysteine protease called sortase and are covalently crosslinked [134,135]. Sortases recognize the C-terminal LPxTG motif together with other important conserved motifs like the pilin motif WxxxVxVYPK, in order to send adhesion proteins to the cell wall or to catalyze inter-molecular cross-linking of proteins for pilus shaft formation [136,137]. Interfering with the recognition of these sequences is another option to disruption cell adhesion. 

Other adhesion mechanisms can be exploited to avoid bacterial infection. Autotransporters, fimbriae and other adhesins are examples of other structures that can be regarded as possible targets for cell adhesion disruption.

#### 3.2.2. Virulence Factors and Toxins

Virulence factors are proteins produced by viruses, bacteria, and other organisms to enable them to colonize a host, create infection, and evade the immune system. In bacteria, virulence factors can be encoded by chromosomal DNA or by plasmids. Different strategies have been studied to counterattack the virulence factors used by bacteria. Several years ago, the concept of therapeutic antibodies appeared, mainly as a treatment for cancer [138]. Since then, research on therapeutic antibacterial antibodies has been giving promising results. So far, only three monoclonal antibodies have been licensed for the treatment of bacterial infections: palivizumab for prevention of respiratory syncytial virus in high-risk infants and raxibacumab and obiltoxaximab for prophylaxis and treatment of anthrax [139]. In general, therapeutic antibacterial antibodies can be divided into two groups: those binding to the pathogen to opsonize it and those binding to toxins or other virulence factors to neutralize them. All of them have narrow specificity and can be produced against any pathogen. Those that bind to virulence factors disarm the bacteria, thereby giving the host a chance to clear the infection immunologically. Therapeutic antibodies against toxins, like shiga toxin, produced by shiga toxin-producing *E. coli*, are currently undergoing clinical trials [140,141]. Obiltoxaximab and raxibacumab, the two antibodies approved for antibacterial prophylaxis, also work by neutralizing anthrax toxin and blocking its entry into host cells [142,143]. As these antibodies have no direct antibacterial activity, the selective pressure for resistance to appear should be low.

#### 3.2.3. Signaling

Bacteria can cause an extensive rewiring of the host signaling pathways during infection. Once bacteria have adhered and colonized the host cells, they use different strategies to enhance their replication and survival. Sometimes, bacteria will hijack the host cell death pathway to allow its replication and other times, they will promote it for replication and spread within the infected organism. The signaling processes during infection are highly regulated. Pathways like the NF-κB–dependent pro-survival, mitochondrial pro-death [144] or the inflammasome-dependent [145] host cell death pathways are usually arrested or modulated by the bacteria in order to permit them to control the infection.

Host cell death is an intrinsic immune defense mechanism in response to microbial infection. Bacterial signaling plays an important role in neutralizing the host immune system so the infection can progress. Cell death pathways like the apoptotic, necrotic, and pyroptotic cell death pathways are usually subverted by the pathogen [146,147,148]. By manipulating cell death, the pathogen can continue to use the host cell to survive, replicate, and to extend the infection to other cells when needed. 

Enteropathogenic *E. coli* (EPEC) has more than 20 effector proteins, all with multiple functions. During infection, the host cell tries to control microbial proliferation and to respond to the infection by the recruitment of immune cells to the site of infection, activating inflammatory pathways and the complement system. When the infection is not rapidly controlled, the host organism induces cell death in the infected cells to terminate the infection [149,150]. It is then when bacterial proteins try to subvert those pathways to the benefit of the bacteria. Such is the case of the bacterial effector protein EspL. When EPEC infects epithelial cells, host adaptor proteins like the receptor-interacting serine/threonine-protein kinase 1 (RIPK1), the receptor-interacting serine/threonine-protein kinase 3 (RIPK3) or the TIR-domain-containing adapter-inducing interferon-β (TRIF) drive a caspase-independent form of cell death termed necroptosis [151,152]. These three proteins contain receptor-interacting protein (RIP) homotypic interaction motifs (RHIMs), which play a key role in cell death and inflammatory signaling [153,154,155]. Enteropathogenic *E. coli* injects EspL into the infected cells using a T3SS. This protein has a cysteine protease activity that cleaves RIPK1, RIPK3, and TRIF in the RHIM motif. This leads to inactivation of inflammatory signaling and contributes to persistent colonization by EPEC [156].

Targeting bacterial proteins that hijack crucial host pathways or targeting the interaction of the bacterial and the host proteins is another alternative for bacterial infection treatment. 

## 4. Another Alternative: Enhancing Host Cell Immune Response

The vast majority of treatments for infections aim to harm the pathogen. However, in this review we have shown how this strategy can lead towards the appearance of resistance. Now, instead of focusing on microbial targets with less potential to create resistance, we want to put the focus on the host immune system. Empowering the host immune system, by recruiting more macrophages to the site of infection or by enabling more efficient bacterial opsonization, could avoid the need of using antibiotics.

Antimicrobial peptides and proteins (AMPs) are evolutionarily conserved molecules of less than 60 amino acids in length [157,158], found in organisms ranging from prokaryotes to humans [159]. They are part of the first-line host defenses found among all classes of life and act as potent, broad-spectrum antibiotics at physiological conditions [160]. Although the principal antibacterial activity is attributed to their membrane-lytic mechanism, AMPs have also been demonstrated to function in host immune modulation, often by enhancing protective immunity and suppressing inflammation. They can influence processes like cytokine production, antigen presentation, chemotaxis [161,162,163,164,165] or wound healing [166,167,168], complementing their bactericidal activity [169]. 

For example, the AMP LL-37, found mainly in the granules of neutrophils, is an antisepsis agent that upregulates the production of interleukin-8 and leads to recruitment of leukocytes to the site of infection [170,171]. The name LL-37 derives from the two leucine (L) residues at the N-terminus and the number of amino acids (37) constituting the peptide. LL-37 activity is sensitive to pH and salt conditions, and it has a broad range of activity against both Gram-negative and Gram-positive bacteria [172]. LL-37′s ability to kill is related to its adoption of an alpha-helical conformation [173], which is favored at physiological pH and in the presence of anions or salt solutions mimicking intracellular fluid. Its alpha-helix disrupts the microbial membrane, causing lysis. However, LL-37 is also involved in other functions such as chemotaxis and apoptosis. Chemotaxis is mediated by the interaction of LL-37 with the Formyl Peptide Receptor Like-1 (FPRL-1), a G-coupled receptor that is found to activate direct migration of immune cells [162,174]. It is a low-affinity peptide–receptor interaction, which means that the interaction will only happen when sufficient LL-37 is in the environment of the receptor, due to release of LL-37 by neutrophils, with the aim of recruiting more immune cells to fight an invasion. LL-37 also inhibits neutrophil cell death and promotes apoptosis in other cell types like smooth muscle or epithelial cells [175,176,177,178], possibly as part of the survival strategy during bacterial infection. To inhibit neutrophil apoptosis, LL-37 interacts with P2X7- and G-protein-coupled receptors (other than FPRL-1), eventually inhibiting the BH3 interacting-domain death agonist (BID) and procaspase-3 cleavage [175,176]. BID is a pro-apoptotic member of the Bcl-2 family involved in the regulation of anti- and pro-apoptotic signals. Procaspase-3 is a member of the cysteine-aspartic acid protease (caspase) family, and once it is activated to caspase-3, it interacts with other caspases as part of the sequential activation cell apoptosis (Figure 7). 

Overall, LL-37 contributes to the enhancement of innate host defenses against acute infection through the regulation of different pathways. One of the advantages that immunomodulation offers is its non-specific nature, suggesting that these AMPs can be used as broad-spectrum protection, enabling prophylactic use in high-risk groups and early treatment before the identification of causative infectious agents [179]. Antimicrobial peptides and proteins with similar characteristics are very promising for the development of potential therapeutics to use against multiple antibiotic-resistant infectious diseases. Aside from the empowering effect in the immune system and their membrane-lytic abilities, AMPs can affect several bacterial processes such as macromolecular synthesis (i.e., RNA, DNA synthesis) [180,181], protease inhibition, and protein-folding inhibition [182,183]. The way AMPs hijack these processes is by peptide–peptide or protein–protein interactions with key molecules in each pathway. Again, AMPs do not create a strong evolutionary pressure towards the development of resistance that conventional antibiotics do [184,185]. Antimicrobial peptides and proteins have been effective in vivo for billions of years. One further benefit might be in AMPs being “dirty” drugs, able to disturb many biological functions simultaneously rather than blocking a specific high-affinity target. 

## 5. Conclusions

Traditional antibiotics kill bacteria by interfering with essential cellular processes. In the last 100 years, antibiotics have been used as the main strategy not only to fight infections but also as prophylaxis in important medical procedures. Antibiotic misuse has led to the current situation where bacterial infections are projected to be the main cause of death by 2050. New antibiotics are continuously being discovered but they face the same drawback: selective pressure towards development of drug resistance is extremely high when essential processes in bacteria are targeted.

In the last decade, approaches aiming to disarm the pathogens instead of killing them have been gaining popularity. Once the pathogen is subdued, it is easier for the host innate immune system to clear up the infection. These approaches aim to block pathogenesis, neutralize toxins or interfere in host–pathogen interactions by targeting protein–protein interactions.

Targeting interactions between the pathogen and the host, should minimize the appearance of resistance. Firstly, targeting bacterial processes not important for cell survival imposes a mild selective pressure towards resistance only. Secondly, the targeted interaction also depends on the host organism, which usually evolves more slowly than bacteria, making host proteins, or their interactions, stable and hence good target candidates. Protein–protein interactions are involved in many decisive processes occurring when there is an infection. Better understanding at the molecular level of processes like quorum sensing, attachment to host membrane by secretion systems or hijacking of the cell death pathway can drive the development of new molecules aiming to block these processes, so that the pathogen cannot establish infection. These new molecules can be refined to overcome the limitations observed with natural sequences such as decreased activity or systemic toxicity [186,187].

Some of the options presented in this review offer good alternatives to the use of antibiotics. Despite the production cost of synthetic peptides and their sometimes low stability in vivo, AMPs, for instance, provide a broad spectrum of effect, as their action is based on their biochemical properties. They could be the perfect candidate to fill in a niche as topical therapeutics toward infection in complicated wounds and ulcers [1]. The production of peptides targeting immunomodulation is also a field that offers vast potential for exploitation, as currently conventional antibiotics are the first line therapeutics used as prophylaxis when an infection can potentially occur. The main advantage that immunomodulation offers is that by targeting the host, rather than the pathogen, it avoids selective pressure for the evolution of microbial resistance. It also has a non-specific nature, suggesting that the designed peptides could be used as broad-spectrum protection against a range of microbial pathogens.

In conclusion, targeting non vital bacterial processes like host–pathogen interactions, cell attachment or immunosuppression, opens up new alternatives to produce drugs that are able to disarm the bacteria or empower the host to avoid disease onset. These alternatives are based in deep knowledge of the affected pathways to be able to develop new molecules targeting protein interactions. The fact that non-essential bacterial processes are targeted, together with the low evolution rate that most host organisms have, will provide a new generation of drugs with a long-lasting life that will hopefully help to overcome the current antibiotics crisis.

## Figures and Tables

**Figure 1 ijms-20-01255-f001:**
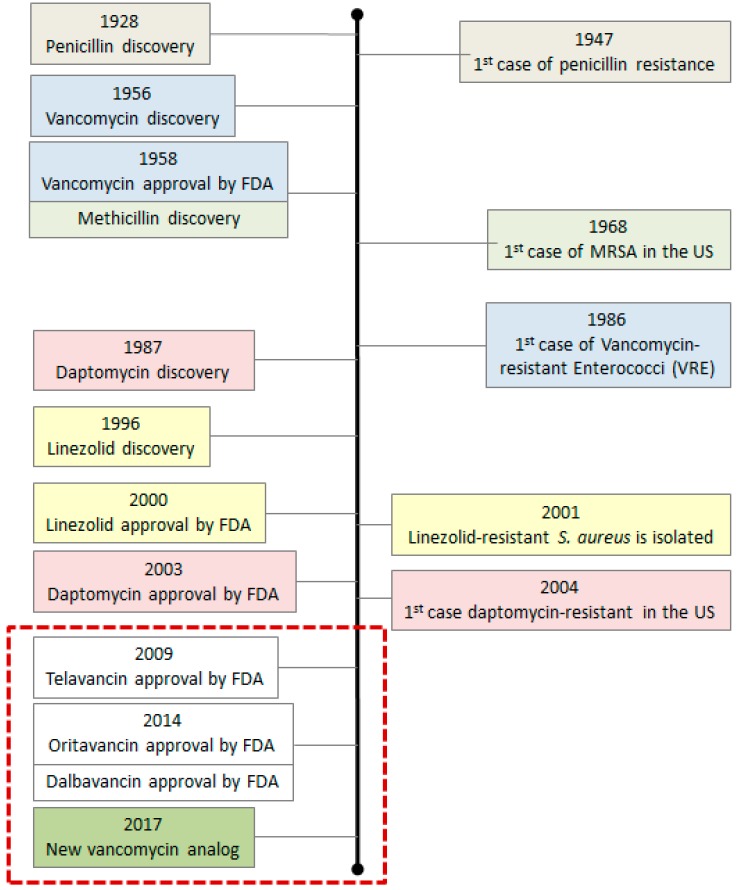
Vancomycin and vancomycin analog’s discovery timeline. **Left side**: antibiotic discovery and approval. **Right side**: appearance of resistance. Each color corresponds to a different antibiotic. The red box shows the last vancomycin analogs approved. Information about methicillin and linezolid is included in the timeline too as they are alternative treatment options available when vancomycin resistance occurs.

**Figure 2 ijms-20-01255-f002:**
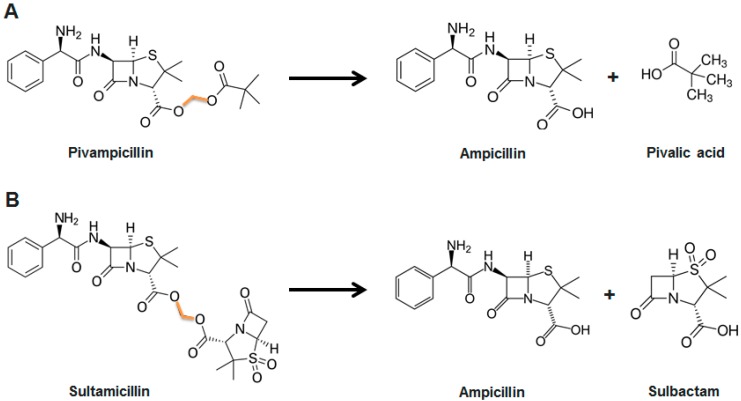
Pivampicillin (**A**) and sultamicillin (**B**) structures. A methylene group (in orange) binds ampicillin with pivalic acid or sulbactam, respectively. When pivampicillin and sultamicillin are processed in the body, ampicillin and pivalic acid/sulbactam are released at equal ratios.

**Figure 3 ijms-20-01255-f003:**
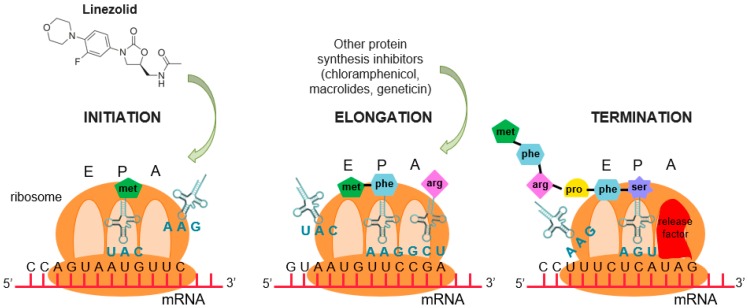
Protein synthesis process. E (exit), P (peptidyl), and A (aminoacyl) correspond to the different sites where the tRNA moves during the elongation phase. Linezolid inhibits the initiation phase of protein synthesis, while other protein synthesis inhibitors interfere in the elongation phase.

**Figure 4 ijms-20-01255-f004:**
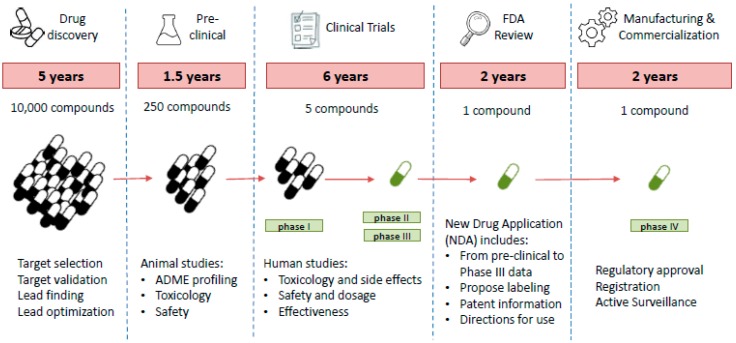
Timeline for antibiotic discovery. Stages for antibiotic discovery and the different steps carried out in each one.

**Figure 5 ijms-20-01255-f005:**
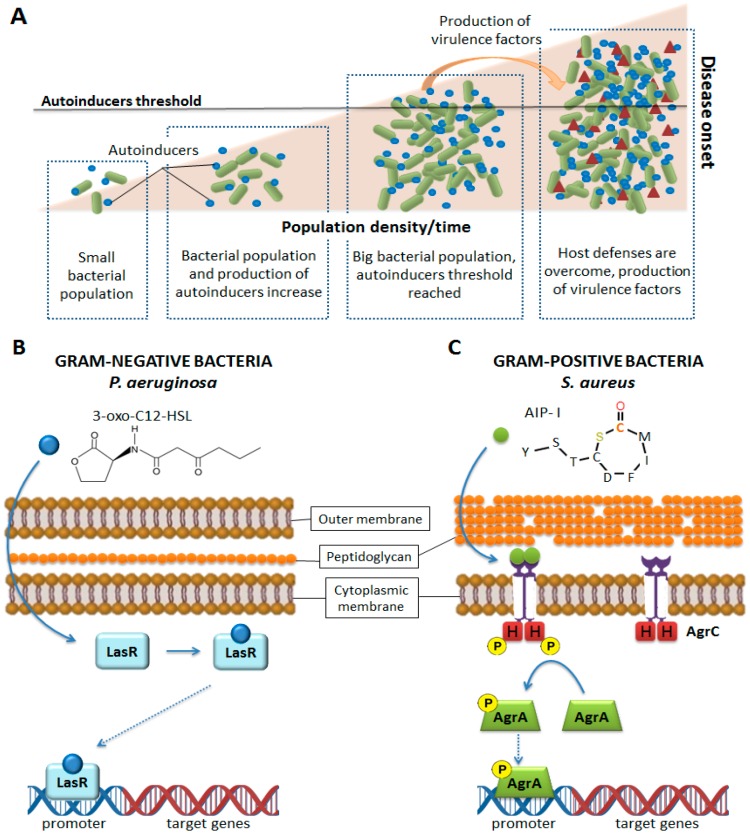
(**A**) Autoinducer production by bacterial. Once a threshold is reached, production of virulence factors, bioluminescence or other molecules takes place. (**B**,**C**) Gram-negative and Gram-positive quorum sensing mechanisms. In Gram-negative bacteria like *Pseudomonas aeruginosa*, the autoinducers freely diffuse into the cell and associate with their receptor. In Gram-positive bacteria like *Staphylococcus aureus*, the receptor is associated with the cell membrane and is phosphorylated upon external binding of the autoinducer.

**Figure 6 ijms-20-01255-f006:**
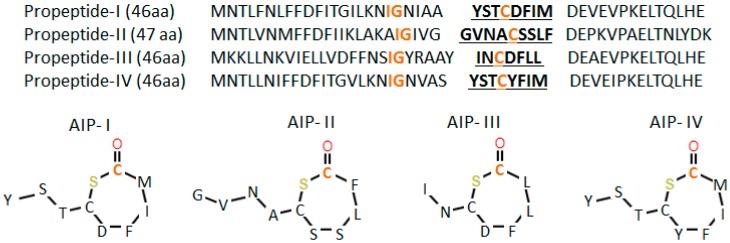
*S. aureus*’s four groups of autoinducer peptides (AIPs) and their propeptide sequences. The orange residues play important roles in propeptide processing. The underlined residues form the AIPs.

**Figure 7 ijms-20-01255-f007:**
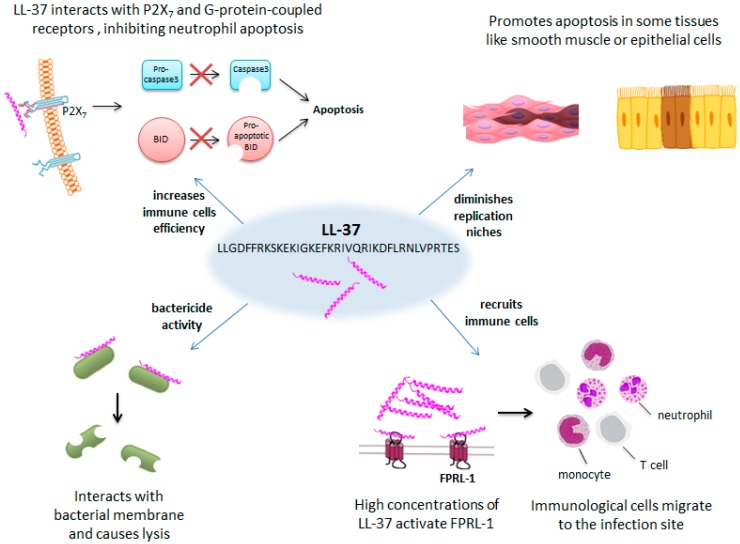
LL-37 antibacterial effect involves multiple strategies. LL-37 causes lysis of the bacterial membrane, attracts immune cell to the infection site, promotes apoptosis in some tissues, and inhibits apoptosis in neutrophils.

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
