# Peer review of "Thinking Outside the Bug: Molecular Targets and Strategies to Overcome Antibiotic Resistance"

_ijms, 2019, doi:10.3390/ijms20061255_

Reviewer 1 Report

I really enjoyed reading this review I found it is a very good summaryabout  where we are in our battles against pathogens. It is comprehensive review without overwhelming the reader. The only thing I would add is more discussion about pathogenic biofilms and what we can do about those super resistant structures.

Author Response

We would like to thank the reviewer for the positive and constructive comments. We have made changes (highlighted in red) in the revised manuscript to address the reviewers queries.

We have added a section on biofilms as follows:

Biofilm formation is a process whereby microorganisms attach and grow on a surface and produce extracellular polymers that facilitate attachment and matrix formation. Biofilms can be formed at the sites of implanted medical devices as implants or catheters[91]. Formation of biofilms around the devices and encrustation is a clinical complication that can threaten the patient’s life. To reduce these infections, systemic antibiotic prophylaxis as well as local administration of antimicrobial agents are administered[1]. Some alternatives being studied are also coating the device with antibacterial compounds[81]or making the implant material resistant to colonizationby physiochemical modification of the biomaterial surface to create anti-adhesive surfaces.This last approach, together with theinhibition of cell-to-cell signaling prior to biofilm formation, resultsespecially interesting as they do not include the use of antibiotics.

Sincerely,

Yann Gambin & Emma Sierecki

Reviewer 2 Report

The manuscript entitled: “Thinking outside the bug: molecular targets and strategies to overcome antibiotic resistance” by Y. Gambin, E. Sierecki and A. Monserrat-Martínez is of interest for the scientific community, especially to those working on the antibiotic field. Overall, the manuscript is well written and contains adequate information.

Before publication, some small things should be considered:

1.-In title and abstract, line 6. The sense of “bug” escapes me. Should it be possible to substitute for some other word?.

2.- Page 4, line 5. “10-8” and “10-9” should be written.

3.- Page 6, lines 7-9. “Hundreds of compounds ……substituting the different fragments  ……modifies its activity” should be better.

4.- Page 13, line 7. Include “that” between “advantages” and “immunomodulation”.

5.- The references sections should be carefully revised and corrected. These are some of the things that I have found:

·         Provide inclusive page numbers with full page range, e.g.  3081-3087 (reference 141).

·         Missing pages should be included in several references, e.g. 34, 35, 44, 49, 67, 70, 82, 86, 91, 92, 94, etc…. among others.

·         Reference 27. “d” should be written instead of “d.”

·         Reference 39. A space should be included between “Rifamycin” and “Mode.”

·         Reference 49. In “b-lactam” symbol should be used for “b” (“β-lactam”).

·         Reference 56. The word “Part” should be removed in the title of the journal.

·         Reference 68. “Semi-Empirical” should be written.

·         Reference 72. Remove one “-“ after “development”.

·         Reference 104. “agr” and “Staphylococcus” should be written in italics.

·         Reference 106. “Staphylococcus epidermidis agr” should be written in italics.

·         Reference 112. “Erwinia carotova” should be written in italics.

·         Missing ending pages should be included in several references, e.g. 138, 156, 178, 187.

Author Response

We would like to thank the reviewer for the positive comments and apologize for the mistakes in referencing. We have corrected all errors and have asked another english native speaker to go over the manuscript.

We hope all the changes made will satisfy the two reviewers,

Sincerely,

Yann Gambin & Emma Sierecki